# New Anti-RSV Nucleoprotein Monoclonal Antibody Pairs Discovered Using Rabbit Phage Display Technology

**DOI:** 10.3390/antib12040073

**Published:** 2023-11-08

**Authors:** Pierre-Emmanuel Baurand, Jérémy Balland, Emilia Galli, Suvi Eklin, Rémy Bruley, Laurence Ringenbach

**Affiliations:** 1Diaclone SAS—Part of Medix Biochemica Group, 6 Rue Dr Jean-François-Xavier Girod, BP 1985, 25000 Besançon, France; 2Medix Biochemica Group, Headquarter, Klovinpellontie 3, FI-02180 Espoo, Finland

**Keywords:** rabbit, phage display, human respiratory syncytial virus, monoclonal antibodies, nucleoprotein, ELISA

## Abstract

Human respiratory syncytial virus (hRSV) is one of the major contagious viruses and causes complicated respiratory issues, especially in young children. The sensitive and fast detection of hRSV is critical for taking the most effective actions. In the present study, rabbit antibodies against the hRSV nucleoprotein (NP) were developed using phage display technology. A female rabbit was immunized with an hRSV strain A2 recombinant NP. A Fab library was built and sorted during two successive panning rounds for strain B and the A2 NP (recombinant preparations), respectively. The choice of candidates was performed using ELISA on the two NP strains. The obtained library was 3 × 10^6^ cfu/mL, with an insertion rate of >95%. The two panning rounds permitted an enrichment factor of 100. ELISA screening allowed us to obtain 28 NP-specific Fab candidates. Among them, 10 retained candidates were reformatted into rabbit full IgG; thereafter, pairing tests on the recombinant strains and native lysate samples were performed. After the pairing tests on the recombinant strains, 53 pairs were identified. Eleven pairs were identified as being able to detect RSVs from native lysates. This work presents new high-potential monoclonal antibodies mAbs (mAbs), which would benefit from lateral flow testing data with patient materials.

## 1. Introduction

Human respiratory syncytial virus (hRSV) is recognized as one of the major contagious viruses that infect the respiratory tract [1,2]. RSV can be serious, especially for infants—it is the most common cause of respiratory hospitalization causing bronchiolitis and pneumonia in children younger than one year [3]. In addition, this virus can be dangerous for adults older than 65 years, especially those with chronic heart or lung disease or those with weak immune systems [2]. RSV can cause significant complications in young children; thus, its diagnosis is very important for taking the most effective actions [4]. Various types of laboratory diagnoses are classically used to test for RSV infection. Among them, two are focused on virus detection in nasopharyngeal swabs: (1) molecular testing by PCR assays focusing on RSV genetic material and (2) antigen testing assays that seek viral fragments. If PCR diagnoses have the advantage of being very sensitive methods, their weakness comes from their lead time for obtaining results (a few hours). The need for specific equipment/infrastructure also leads to PCR being inaccessible, particularly in economically deprived countries [4]. Rapid diagnosis tests based on antigen detection with mAbs have been highlighted during the recent SARS-CoV-2 pandemic. Their main advantages come from their short time for result delivery (15 min) and their affordability. The downside is that their sensitivity and specificity are often lower than those of PCR testing [4], and these elements need to be improved.

The fusion (F) proteins and nucleoproteins (NPs) of viruses are two major targets used for antigen detection. Concerning hRSV, F proteins are more studied than NPs as they interact with larger numbers of neutralizing existing mAbs due to their therapeutic potential [5,6]. One advantage of using NPs as targets for diagnostic mAbs is their high level of conservation. NPs have been largely used in sensitive diagnostic tests related to influenza [7] and SARS-CoV-2 [8]. Although NPs are the most abundant components of viruses leading to the highest immune responses during infection, surprisingly, studies on mAbs against the hRSV-NP are rare. Few publications focusing on mAb development against hRSV-NP are available, though various mouse mAbs have been used in different approaches including ELISA [9], and recently, they have been used in rapid fluorescent immunochromatographic strip tests [10]. These new mAbs are useful tools available for the diagnosis and detection of RSV infections in clinical and research samples [9].

Rabbits, due to their distinct evolutionary path from rodents [11], are a good and interesting alternative species for mAb development, especially for weakly immunogenic antigens in mice [12]. Rabbit mAbs have generally higher avidity, binding affinity, and specificity against various types of antigens compared to mice mAbs [13,14]. Classical rabbit mAb generation is complicated due to the instability of rabbit hybridomas and decreases in secretion over time [11]. Phage display, which is a powerful recombinant technology that allows for the exposition of small antibody fragments (generally Fabs or scFvs) against a target of interest on a phage’s surface, appears to be more adapted to rabbit mAb development using a Fab chimeric format [15]. Recently, the potential of rabbits as good models for mAb development against viruses has been highlighted by the discovery of rabbit antibodies against the SARS-CoV-2 spike protein [16]. Although rabbit mAbs have proven their potential for in vitro diagnostics purposes, until now, there have been no rabbit mAbs against hRSV-NP available on the market. In this context, the goal of our study was to develop specific anti-hRSV-NP mAbs using rabbit phage display technology. The new rabbit mAbs generated could be used as raw materials for setting up sensitive diagnosis tests targeting the hRSV-NP.

## 2. Materials and Methods

### 2.1. Immunization

The rabbit immunization was outsourced (Covalab company, Dijon, France). One female New Zealand White weighing 2.5 kg (Oryctolagus cuniculus cuniculus) was immunized using a 42-day protocol with recombinant the hRSV nucleoprotein, strain A2 (reference 40821-V08E, Sinobiological Europe GmbH, Eschborn, Germany). Briefly, 3 weekly subcutaneous injections of 200 µg of antigen mixed with incomplete Freund adjuvant were performed on days 0, 7, and 14. On day 34, a final boost by intravenous injection was performed with 200 µg of antigen. Serum samples were taken on days 0, 28, and 42 to determine the global immune response. The specificity of the serum along the immunization protocol was tested by ELISA on 50 ng of recombinant antigens per well (the A2 and B strains) coated overnight at 4 °C. The serum dilutions were performed from 1/100 to 1/409,600 on 7 points in PBS. Anti-rabbit IgG HRP (reference: BI2407, Abliance, Compiègne, France) was used for the detection. The negative control was performed using PBS. The OD_450nm_ was measured with a BioTek ELx808 absorbance plate reader (Agilent, Santa Clara, CA, USA). On day 42, the spleen was collected and used for the subsequent molecular biology steps (RNA isolation, cDNA synthesis, and library building).

### 2.2. Library Construction

The total RNA isolation was performed starting with the spleen cells (6 × 10^8^) with a Rneasy Maxi kit (Qiagen, Hilden, Germany) following the manufacturer’s instructions. The RNA extract was converted into cDNA with SuperScript III Reverse Transcriptase (Invitrogen, Waltham, MA, USA) according to the manufacturer’s instructions.

A chimeric Fab library (the variable part from the rabbit and the constant Fab part from the mouse Kappa/IgG1 domain) was built in our proprietary phagemid [17]. The histidine tag was followed by a Myctag in the C-terminal part of the construction (Figure 1).

The rabbit variable IgG VH and VL sequences were amplified following 2 rounds of successive PCRs using a set of primers adapted from Nguyen et al. [18]. The PCR protocol was a modification of the method used by Baurand et al. [17]. The first PCR rounds were performed to amplify the VH or VL sequences starting from the cDNA. The amplifications were performed with Phusion DNA Polymerase (Thermo Fisher Scientific, Waltham, MA, USA) using the following PCR program: 94 °C, 2 min; 30 cycles: 94 °C for 30 s, 60 °C for 30 s, 72 °C for 60 s; and 72 °C, 7 min (the final amplification). The second round of PCRs was performed on the purified first-round PCR products using the following PCR program: 94 °C, 2 min; 25 cycles: 94 °C for 30 s, 62 °C for 30 s, 72 °C for 60 s; and 72 °C, 7 min (the final amplification). This second PCR round served to add 20 bp of recombination parts to the extremities of the VH/VL fragments amplified from the first PCR round. For the VH PCRs, 5 different forward and reverse primer pairs were used. For the VL PCRs, 9 different forward and reverse primer pairs were used (8 for kappa and 1 for lambda). The complete chimeric Fab was built by recombination cloning (NEBuilder^®^, New England Biolabs, Ipswich, MA, USA). The purified final Fab products (~1500 bp) were cloned in our phagemid vector between the *ApaL*I and *Not*I enzyme restriction sites. The library was finally constructed by electroporation of the cloned phagemids into electrocompetent TG1 cells (Lucigen, Middeleton, WI, USA).

### 2.3. Library Validation

Validation was performed using the entire protocol described by Baurand et al. [17]. Five µL spots of the 10–10^6^ diluted library were applied to LB agar plates and incubated overnight at 37 °C. The next day, the library size was determined by counting the colonies in the spots. The PCR product insertions (in percentages) in the phagemid were evaluated by the colony PCRs on 48 randomly chosen clones. The expected sizes for the positive inserts were near 1800 bp. To confirm the proper insertions and reading frames in the phagemid, a sequencing control was made using 8 independent positive clones.

### 2.4. Phage Infection and Preparation

The final library was cultured and super-infected using the helper phage M13K07 (Invitrogen, Waltham, MA, USA). After overnight culture in LB agar media supplemented with 2% glucose (at 37 °C, 200 rpm), the Fab phage library was precipitated by polyethylene glycol (PEG6000 20%/2.5 M NaCl). The purified phages were resuspended in 1 mL of sterile PBS [18].

### 2.5. Phage Display

#### 2.5.1. Panning Selection

Two successive panning rounds were performed on the phage library. The protocol used was adapted from that described by Russo et al. [19]. The first selection round was performed on 2 independent Maxisorp microtiter plate (NUNC 439454) wells (w) that were coated overnight in PBS at 4 °C with 0.1 and 1 µg/w of recombinant NP, strain B (reference 40822-V07E, Sinobiological Europe GmbH, Eschborn, Germany). A control well without protein (condition “0”) was performed in parallel. Ten μL of the Fab phage library was incubated for 60 min at RT with slow shaking. Then, the wells were washed 5 times with PBS-Tween 0.5% buffer. Finally, the phages retained on the NP-coated wells were eluted with trypsin.

The second round of panning was performed with the same scheme used in the first round, except that recombinant NPs from strain A were used. Two µL of the Fab phage library from the first round were incubated for 60 min at RT with slow shaking. Then, the wells were washed 20 times with 0.5% PBS-Tween buffer. Finally, the phages retained were eluted with trypsin. After each round of biopanning, a new TG1 infection was performed, and the phages were incubated overnight. Spots of 5 µL were performed on the LB agar plates to determine the enrichment of the library between the input (before panning) and the output (after panning). The output/input ratio was calculated [18].

#### 2.5.2. Screening

Ninety-four randomly selected Fab clones from the second round of panning were set up overnight after IPTG addition to induce periplasmic expression in 1 mL of LB in deep plate-wells. Additionally, one positive control was established with rabbit serum diluted 500 times, and one negative control with a periplasmic extract from empty bacteria was set up. Periplasmic extracts containing Fab clones were screened by ELISA for the detection of the two strains’ NPs coated on the microtiter wells. For detection, a rabbit anti-Myc-HRP antibody (reference A190-105P, Bethyl Laboratories, Montgomery, TX, USA) was used. In parallel, a negative control was set up with an irrelevant antigen (SARS-CoV-2, recombinant receptor binding domain, reference 715-H17-0BU, Diaclone SAS, Besançon, France). The OD_450nm_ was measured using a BioTek ELx808 absorbance plate reader (Agilent, Santa Clara, CA, USA).

### 2.6. Recombinant Engineering

#### 2.6.1. Identification of Candidates

Based on the OD measurements obtained from the ELISA Fab screening, the best 48 candidates were analyzed by sequencing to determine their nucleotide and amino acid sequences and to eliminate the redundant clones. Based on the CDR3 VH and VL domain analyses, the clones were classified into various families.

#### 2.6.2. Reformatting and Recombinant Production of the Full IgG

The selected Fab candidates were reformatted in full rabbit IgG form. The full rabbit IgG heavy constant part used for reformatting was amplified by the PCR from the cDNA generated for the library building with specific primers (5′ to 3′) for Rab-start-VH: GGGCAACCTAAGGCTCCA (forward) and Rab-end-VH: CTATTTACCCGGAGAGCGGGA (reverse). The 969 bp amplified product was sequenced (Microsynth France SAS, Lyon, France). The translated 323 amino acid sequence was compared to databases (Protein BLAST: search protein databases using a protein query (nih.gov), accessed on 12 December 2022), and it was 100% identical to the rabbit Ig heavy chain (accession number AKM12456.1). The kappa constant part was amplified by the PCR from the cDNA with specific primers (5′ to 3′) for Rab-VK4: GGTGATCCAGTTGCACCT (forward) and Rab-end-K4: GAGGTCGAGGTCGGGGGATCCTTATCACTAACAGTCACCCCTATTG (reverse). The 309 bp amplified product was sequenced (Microsynth France SAS, Lyon, France). This kappa sequence coding a 103 amino acid chain was identified as idiotype rabbit kappa b4 (100% identical to sequence ID P01840.1).

To produce the full rabbit IgG for testing purposes, the complete DNA chains (the variable plus constant domains) of the chosen candidates were subcloned in licensed monocistronic mammalian expression vector (one for the light and one for the heavy chain, as shown in Figure 2). The transient production of the IgGs in the CHO cells was outsourced to RD-Biotech company, Besançon, France. In brief, 50 mL of suspension CHO cells was transfected with the plasmids, and the supernatants were collected 14 days post-transfection.

The transfection supernatants were purified on a 1 mL HiTrap MAbselect PrismA protein A column (reference 17549851, Cytiva, Marlborough, MA, USA). The purified antibody concentration in mg/mL was determined using a Nanodrop 2000 (Thermo Fisher Scientific, Waltham, MA, USA) with an extinction coefficient of 1.4. The quality of the purified rabbit IgG was visually assessed after electrophoresis on SDS-PAGE gel 4–15%.

### 2.7. Pairing Tests

#### 2.7.1. Recombinant Strains

After an individual specificity validation on the coated NP-hRSV recombinant strains, all of the purified recombinant full IgGs were tested in pairs using sandwich ELISA. The 2 recombinant NPs from strains A2 and B were used as the antigens in the pair assays. Each of the candidates was tested as capture (coated in a microplate) and detection (biotinylated) antibodies using the following protocol: the antibodies were coated at 1 µg/w in PBS overnight at 4 °C. The day after, the wells were blocked over 2 h with 1% BSA-PBS and 10% sucrose. A dilution series of recombinant antigens ranging from 5 to 0.2 ng/w with a dilution factor of one-half was made. The antigens were incubated for 2 h at RT, followed by 4 washes (0.05%PBS-Tween, 300 µL/w). Then, secondary biotinylated antibodies (1 µg/mL) were added to the wells and incubated for 1 h at RT, followed by 4 washes (0.05%PBS-Tween, 300 µL/w). Finally, 100 µg/mL of streptavidin-HRP diluted at 100 µL/w was added to wells for 30 min of incubation at RT, and then 4 washes (0.05%PBS-Tween, 300 µL/w) were performed. Tetramethylbenzidine at 100 µL/w (TMB) was added to wells over 15 min, and the reaction was stopped using 1 N H_2_SO_4_ at 100 µL/w. The OD_450nm_ was measured with a BioTek ELx808 absorbance plate reader (Agilent, Santa Clara, CA, USA). To explore if the pairs did not recognize the NPs from other viruses, positive pairs were tested additionally with recombinant SARS-CoV-2 NPs (reference 715-H17-0BU, Diaclone SAS, Besançon, France).

#### 2.7.2. Native Lysates

The positive pairs identified by the assays using recombinant antigens were tested on native supernatant lysates (SNs) (Medix Biochemica, Espoo, Finland) produced by infecting Hep-2 cells in vitro (catalog number: FC517, Respiratory Syncytial Virus (RSV) Antigen, Native (medixbiochemica.com, accessed on 5 January 2023)). The same pairing protocol as described in Section 2.6.1 was used, with the exception that the native antigen lysate preparation was added in dilutions ranging from 1:20 to 1:960. The dilutions were prepared in PBS. Additionally, a blank sample containing only PBS was added to all antibody pairs. The OD_450nm_ was measured with a BioTek ELx808 absorbance plate reader (Agilent, Santa Clara, CA, USA).

## 3. Results

### 3.1. Immunization

The rabbit sera were tested on days 0, 28 (mid-time), and 42 (end) using ELISA on the two coated recombinant NP strains (50 ng/w).

As shown in Figure 1, specific immune responses were detected on days 28 and 42 and compared to the day 0 sample. The immune responses were globally the same against the two strains tested on days 28 and 42. A strong signal (OD of >1) was detected until 102,400 dilutions of the serum.

### 3.2. Library Construction

The quality of the chimeric Fab fragments obtained after the PCR and recombination steps was assessed by 1% agarose (*w*/*v*) gel electrophoresis in a TAE 1X buffer (30 min at 135 V). As shown in Figure 2, clear chimeric Fab VH and VL fragments near 800 bp were obtained. After an agarose extraction step using the kit protocol for Nucleospin PCR and Clean-up (Macherey-Nagel, Oensingen, Germany), these two chimeric fragments were used in recombination reactions to obtain a full Fab final insert (VH plus VL). After electrophoresis, a clear and proper 1600 bp fragment (without visible contaminant bands of <1600 bp) was visualized on 1% agarose gel (*w*/*v*; TAE 1X) (Figure 2).

The quality of the generated Fab library was evaluated against the following three criteria, as established by Nguyen et al. [18], though with some adaptations:-The library size was to be >10^6^ colony-forming units per mL (cfu/mL).-The insertion rate (IR) was determined after the colony PCRs on the randomly chosen clones from the bacterial spread of the electroporated library. A clone was qualified as positive when it showed a 1800 bp PCR product. A library with good quality should show >80% IR.-The quality of the PCR inserts cloned in our phagemid was assessed by sequencing eight randomly chosen positive PCR products to determine the percentage of plasmids with an ORF for the chimeric Fab and for the evaluation of the sequence variability. A library with good diversity should show <10% identical sequences.

As shown in Table 1, all three pre-set quality criteria were fulfilled. The obtained library size was 3.6 × 10^6^ cfu/mL, with a very high level of insertion (>95% of positive inserts or 46/48 clones). All eight Fab sequences analyzed showed correct ORF-encoding complete chimeric Fabs, and so the number of sequences presenting a Fab ORF was estimated as >95%. In the present study, no redundance was found for the analyzed clones. The diversity of the sampling was maximal.

### 3.3. Panning Selection

Ratios of retention of 5 × 10^−7^ were observed after the first panning round (on the B strain) and 10^−5^ after the second panning round on the A2 strain (Table 1). The titer number obtained after the first rounds was 4 × 10^4^ pfu and 2 × 10^5^ pfu after 2 rounds. These 2 panning rounds lead to an enrichment factor of 100 (from a ratio of 10^−7^ to 10^−5^). This factor was equal to our standard of 100, which demonstrated that no further panning rounds were necessary.

### 3.4. Fab Screening by ELISA

After the biopanning rounds of selection, the next step was to identify the individual Fab clones recognizing both the A2 and B strains’ recombinant NPs. ELISA on coated antigens was performed on 94 individual Fab periplasmic extracts.

Figure 3 shows the mean OD_450nm_ values obtained for the 94 individual Fabs tested for the detection of the recombinant NPs, strains A2 and B, plus an irrelevant protein (IP). Firstly, a low background was obtained for the IP, with a maximum OD_450nm_ value of 0.188 for the H1 sample and a mean OD value of 0.102 for the 94 samples. As shown in the table, more than half of the screened Fab candidates provided positive signals for the two strains’ NPs, with OD_450nm_ values of >0.7 (corresponding to more than three times the maximum background (0.188) obtained for the IP). In detail, 60 Fabs provided OD_450nm_ values of >0.7 for the A2 strain, whereas 57 clones provided OD_450nm_ values of >0.7 for the B strain. In total, 56 individual Fab clones recognized both strains.

### 3.5. Identification of the Fab Clones

Based on the ELISA results, 48 Fabs recognizing both strains’ NPs were sequenced to identify their nucleotide and amino acid sequences and to eliminate the redundant clones. The retained clones are listed in Table 2.

After the analysis, 28 and 24 unique variable heavy and light sequences, respectively, were identified (Table 3).

Based on the CDR3 VH and VL analyses, the candidates were classified into various families [18]. As shown in Table 3 and Figure 3, 13 and 11 families were identified for the CDR3 VH and VL, respectively.

For the subsequent full IgG reformatting step, one sequence from each family was chosen, resulting in ten different sequences. Figure 4 shows the names of the selected candidates and the mean OD values measured as the reactivity with both recombinant strains (A2 and B).

### 3.6. Reformatting and Production of the Full Recombinant Rabbit IgG

After reformatting the Fabs into full rabbit IgG clones, pilot batches were produced over 14 days in 50 mL of the transfected CHO cells. After production, the 10 full IgGs were purified on protein A. As shown in Table 4, between 1.9 and 8.2 mg of recombinant rabbit mAbs were obtained from the transient transfections in CHO cells. The best production yield was obtained for the F12 candidate, with 164 µg/mL.

The quality of the purified rabbit IgG was visually assessed after electrophoresis on SDS page gel 4–15%. As shown in Figure 5, the purity was high (>85%).

The theoretical molecular weight range for the rabbit VH full chains was 47.8–49 kDa, and that for the rabbit VL chains was 22.3–22.7 kDa. The full IgG expected sizes should be near the theoretically determined 140–143 kDa.

As shown in Figure 5, for the three candidates presented here, in reduced conditions, the fragments identified near 25 kDa were VL chains, and those near 50 kDa were VH chains. In non-reduced conditions, a significant majority of the IgG full forms were detected with sizes near 150 kDa.

### 3.7. Pairing Tests

#### 3.7.1. Recombinant Samples

In total, 100 pairs were tested on recombinant strains: each of the 10 candidates was tested as a capture mAb and detector mAb with itself and the 9 remaining candidates. Among the 100 pairs tested, 53 of them were identified as being able to detect recombinant NPs from strains A2 and B (Figure 6). The positive pairs were tested against SARS-CoV-2 recombinant NPs to ensure their specificity. No signal was obtained during our ELISA sandwich protocol.

As shown in Figure 6, the 10 candidates used as capture mAbs were able to make pairs with the three detection mAbs (G1, G10, and G11-biot). Conversely, G1 and G11, as capture mAbs, also paired with all 10 detection mAbs (biotinylated). The raw data are available in the Appendix A.

At a medium level, G10 paired with itself, and A9 capture mAb pairs with the detection mAbs A12, C11, E11, F12, A9, and A10-biot. D11 capture mAb pairs with C11, and E11 and F12-biot capture candidates with lower signals.

#### 3.7.2. Native Lysates

Thirty pairs that were chosen based on testing with the recombinant NPs were further tested with the RSV native lysate preparation (Figure 7). The three biotinylated candidates G1, G10, and G11 were used as detection antibodies, and they paired with all 10 of the candidates used as capture antibodies.

This second test revealed 17 pairs that were able to detect the native RSV lysate (noted as “+++” and “++” in Figure 7). Among them, 11 very good potential pairs that could recognize the native lysate were identified (OD = 5 for two or three of the seven points in the dilution range, as noted in dark grey in Figure 7). Two candidates from the three detection ones tested (G1 and G11) appeared to have the best potential to be used as detection mAbs for the sandwich ELISA. Candidate G1 provided the best results for detection by pairing with 7 of the 10 captured clones (A9, A10, A12, C11, E11, D11, and F12). Candidate G11 had very good results with three capture candidates (A9, A10, and A12) and medium results with four other ones (“++” in Figure 7). In contrast, G10, when used as a detection mAb, had the worst results. Based on our test, seven candidates showed very satisfactory results as capture mAbs by pairing with the G1 and G11 detection mAbs: A09, A10, A12, C11, D11, E11, and F12. The best pair identified in our tested conditions was the couple A09/G1-Biot, with three first points of dilution ranging within OD = 5 and all points of dilution within OD >1 (Figure 8). The raw data are available in the Appendix A.

## 4. Discussion

Our work describes the successful development of several rabbit mAbs against NP-hRSV using phage display technology. Rabbits appear to be a very interesting species model for significant affinity mAb development.

Firstly, the quality of our generated Fab library was a real success, with very high insertion (>95%) and an excellent size (3.6 × 10^6^ cfu/mL corresponding to 1.5 × 10^7^ transformant). These criteria were in the same range as those in our previous work with mouse scFv [17] and those in a rabbit/human chimeric Fab library against the human A33 antigen where Rader et al. [20] reported a size of 2 × 10^7^ transformants. The overall library quality was reinforced by the results obtained in the ELISA screening on the NP recombinant strains, where more than 50% of the tested candidates had positive and specific signals.

Secondly, our library with a chimeric Fab design (rabbit/mouse) was rare as, to our knowledge, it was the first time this design was chosen and published. Usually, a human backbone is preferred in cases of therapeutic development, but for the generation of diagnostic mAbs, we chose to work on a mouse IgG constant part as we knew that it would work well in our phage display process. Most of the past research teams working on rabbit phage display generally used a rabbit/human chimeric Fab construction [18,20,21] or even an scFv format [22]. The main advantage of using a natural Fab is its robust monomeric nature that permits affinity-driven selections compared to the “unnatural” scFv and its multimeric tendency [11]. As we were looking for full IgG candidates, we preferred working on a Fab to maximize the conservation of the binding activities of all of the immunoglobulin antibodies [18].

Thirdly, the panning selection was a real success. Only two rounds on recombinant NP strains were required to obtain good enrichment of specific phages in the library. We had already obtained good results with biopanning on cells in our recent work [17], but with recombinant antigens, we had never obtained such results in only two rounds (generally, three or four rounds are necessary).

Additionally, after the screening steps and the subsequent positive candidate sequencing, 24 candidates were identified, which corresponded to one-quarter of the total number of screened candidates. This was the first time we obtained this magnitude of results after an ELISA screen. Generally, between 1 and 8 candidates were identified (corresponding to less than 12% of screened candidates). This point can be explained by the fact the target was part of a virus which is known to induce a significant immune response. In addition, the quality of the antigens used and the known high binding affinity and specificity of rabbit mAbs against the antigens (compared to mice [13,14]) could explain the good achievement of this part. In comparison, Rader et al. [20] performed four panning rounds to select 40 clones and finally identified three distinct candidates in their work on a chimeric rabbit/human Fab naïve library against the human A33 antigen.

Fourthly, the present work was, to our knowledge, the first to use rabbits for the development of mAbs against NP-hRSV. The results of the present work demonstrate the huge potential of rabbits as interesting species for mAb development against viruses for diagnostic purposes. This potential had been already highlighted by Makdasi et al. [16] in their work on SARS-CoV-2. In addition, our recombinant strategy was successful. We decided to work with rabbit full recombinant mAbs to increase our chances of obtaining the best affinity binders near a natural rabbit mAbs format. We achieved the identification of several antigen-specific chimeric (rabbit/mouse) Fabs and were able to retain their activities when reformatting them into full rabbit IgG formats. The switch of the constant part from mouse to rabbit in the full IgG format was implemented to avoid folding issues as there are different cys bridges between mouse and rabbit mAbs. The rabbit recombinant mAb production was a success, with the same production yield that we have routinely obtained with mouse mAbs.

Finally, in this work, we described the development of several mAb candidates for the detection of RSV NPs. The functionality of these antibodies in a final diagnostic assay setup and with patient materials still needs to be demonstrated. Tien et al. [10] had already highlighted the potential of using mouse mAbs against NP-hRSV for diagnosing infection using a rapid fluorescent immunochromatographic strip test (FICT). In our study, the 11 rabbit mAb pairs we discovered that recognized the native RSV lysate constitute new raw materials with diagnosis applications, even if further validations are required to ensure their abilities to be used in lateral flow antigen detection tests using human nasopharyngeal swabs. González et al. [4] had already demonstrated the potential of mouse mAb pairs against P-hRSV for detecting specifically less than 50 ng of antigen in biological samples. After more sensitivity studies using rapid antigen detection tests (RADTs), our 10 NP-hRSV-specific rabbit mAbs may constitute potential candidates for attempting to increase the known limited sensitivity (~60% to 89%) of this type of test compared with the RT-PCR method mentioned by Pfeil et al. [23]

## 5. Conclusions

This work permitted the identification of several NP-hRSV-specific recombinant rabbit mAbs. After recombinant production of 10 selected candidates, 100 pairing tests were set up on recombinant RSV strains, leading to the selection of 3 detection and 10 capture mAbs. These 13 retained mAbs, constituting 30 pairs, were tested on hRSV lysates. This second test highlighted 11 very good potential pairs able to specifically detect NP-hRSV in native samples. This work presents new raw materials with diagnosis applications.

## Data Availability

All related data and methods are presented in this paper.

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
