# Peer review of "New Anti-RSV Nucleoprotein Monoclonal Antibody Pairs Discovered Using Rabbit Phage Display Technology"

_2073-4468, 2023, doi:10.3390/antib12040073_

Round 1
Reviewer 1 Report
Comments and Suggestions for Authors
The manuscript by Baurand et al. entitled “New anti-RSV nucleoprotein monoclonal antibody pairs discovered using rabbit phage display technology” evaluated. By testing serology and respiratory panels to find the pathogen involved in the infectious process.
The authors discussed the test results and concluded that undiagnosed mumps may be due to a lack of clear diagnostic algorithms. However, there are still many minor problems in the manuscript, which need further revision and improvement. The specific amendments are as follows:
1、 The language needs considerable attention. Try to minimize the use of words such as “We developed”, “We observed” et al. in the article.
2、 In the Abstract, “an” should be changed “a” in line 15. In line 19, unit “ml” should be written as “mL”. In line 23, what is “mabs”? It needs to be explained.
3、 In the Introduction, “for” should be changed “in” in line 43. In line55, “highest” needs to be preceded by “the”. In line 65, “type” should be changed “types”.
4、 In the Materials and Methods, “at” should be changed “on” in lines 86-87. In line 99, “manufacturer’s” needs to be preceded by “the”. In line 114, format error, it needs to be changed to indent the first line by two characters. In line 122, “72” has an extra space in the middle. In line 129, “ApaLI and NotI” should be changed “ApaL I and Not I”. In lines 234 and 241, “were” should be changed “was”. In line 252, “umber” should be changed “number”.
5、 In the Results, “at” should be changed “on” in lines 263 and 265. In line 292, “with” should be deleted. In line 318, “were” should be changed “was”. In line 343, format error, it needs to be changed to indent the first line by two characters. In line 364, unit “ml” should be written as “mL”. In line 431, “see” should be deleted.
6、 In Figure 1, the clarity of the image needs to be improved. In Tables 2 and 5, what does the shaded part refer to? Need to be identified.
7、 In the Discussion, the “CFU” is uppercase, while in the previous description, cfu is lowercase. Please unify before and after in line 445. In line 469, “various” should be deleted. In line 478, “rabbit” should be changed “rabbits”. In line 482, “with” should be changed “in”.
8、 There are many results in the article, and a conclusion section needs to be added to briefly summarize the article.
9、 In the References, the sixth, eighth references lack a page number.
Author Response
Reviewer 1
The manuscript by Baurand et al. entitled “New anti-RSV nucleoprotein monoclonal antibody pairs discovered using rabbit phage display technology” evaluated. By testing serology and respiratory panels to find the pathogen involved in the infectious process.
The authors discussed the test results and concluded that undiagnosed mumps may be due to a lack of clear diagnostic algorithms. However, there are still many minor problems in the manuscript, which need further revision and improvement. The specific amendments are as follows:
1、 The language needs considerable attention. Try to minimize the use of words such as “We developed”, “We observed” et al. in the article.
Changes made in the text
2、 In the Abstract, “an” should be changed “a” in line 15. In line 19, unit “ml” should be written as “mL”. In line 23, what is “mabs”? It needs to be explained.
Changes made in the text
3、 In the Introduction, “for” should be changed “in” in line 43. In line55, “highest” needs to be preceded by “the”. In line 65, “type” should be changed “types”.
Changes made in the text
4、 In the Materials and Methods, “at” should be changed “on” in lines 86-87. In line 99, “manufacturer’s” needs to be preceded by “the”. In line 114, format error, it needs to be changed to indent the first line by two characters. In line 122, “72” has an extra space in the middle. In line 129, “ApaLI and NotI” should be changed “ApaL I and Not I”. In lines 234 and 241, “were” should be changed “was”. In line 252, “umber” should be changed “number”.
Changes made in the text
5、 In the Results, “at” should be changed “on” in lines 263 and 265. In line 292, “with” should be deleted. In line 318, “were” should be changed “was”. In line 343, format error, it needs to be changed to indent the first line by two characters. In line 364, unit “ml” should be written as “mL”. In line 431, “see” should be deleted.
Changes made in the text
6、 In Figure 1, the clarity of the image needs to be improved. In Tables 2 and 5, what does the shaded part refer to? Need to be identified.
Figure 1 has been improved to be more visible.
Clarifications has been added in Table 2 and 5 legends to explain the shaded part
7、 In the Discussion, the “CFU” is uppercase, while in the previous description, cfu is lowercase. Please unify before and after in line 445. In line 469, “various” should be deleted. In line 478, “rabbit” should be changed “rabbits”. In line 482, “with” should be changed “in”.
Changes made in the text
8、 There are many results in the article, and a conclusion section needs to be added to briefly summarize the article.
Conclusion section has been added at the end of the manuscript.
9、 In the References, the sixth, eighth references lack a page number.
Page numbers are now added.
Reviewer 2 Report
Comments and Suggestions for Authors
The study leverages rabbit phage display technology to elucidate the development of novel anti-respiratory syncytial virus (RSV) nucleoprotein monoclonal antibodies (mAbs). A total of 11 pairs, capable of detecting RSV from native lysates, were identified. Nonetheless, several aspects necessitate further exploration.
In Line 347, it would be advantageous for the author to elaborate on how the candidates were classified into different families, and to furnish additional insights, such as, but not limited to, the sequence alignments, clustering, or phylogenetic analysis.
Subsequent to several rounds of pairing tests, the selection was refined to 10 candidates. It is recommended to include a thorough sequence alignment analysis to identify any unique or common motifs across these candidates. Moreover, examining the interface between the antibody and the antigen—either by comparing with known structures or utilizing predictions from AlphaFold—could potentially augment the understanding of binding mechanisms, thereby enhancing the significance of the study.
Author Response
Reviewer 2
The study leverages rabbit phage display technology to elucidate the development of novel anti-respiratory syncytial virus (RSV) nucleoprotein monoclonal antibodies (mAbs). A total of 11 pairs, capable of detecting RSV from native lysates, were identified. Nonetheless, several aspects necessitate further exploration.
In Line 347, it would be advantageous for the author to elaborate on how the candidates were classified into different families, and to furnish additional insights, such as, but not limited to, the sequence alignments, clustering, or phylogenetic analysis.
Subsequent to several rounds of pairing tests, the selection was refined to 10 candidates. It is recommended to include a thorough sequence alignment analysis to identify any unique or common motifs across these candidates. Moreover, examining the interface between the antibody and the antigen—either by comparing with known structures or utilizing predictions from AlphaFold—could potentially augment the understanding of binding mechanisms, thereby enhancing the significance of the study.
A new Scheme 3 showing VH-CDR3 alignments representing the eleven identified families has been added to the manuscript. 3D structure analysis will be for sure very helpful to more investigate the work done in our present study, but unfortunately, this type of analysis is very expansive.
Reviewer 3 Report
Comments and Suggestions for Authors
The authors describe the development of rabbit antibody pairs for diagnostic purposes, namely for sensitive detection of NP protein of hRSV. After rabbit immunization, a phage library displaying rabbit-mouse chimeric Fab was created and then selected for antigen binding. Several clones were discovered and reformatted into full-length rabbit antibodies. They then tested which antibodies work well in pairs as a capture and detection reagents, and performed a titration to discover the threshold of detection sensitivity.
The purpose and the structure of the study are clearly explained and the inference behind the experiments is clear. The presentation of data could be better (please see the remarks below). Importantly, at least for the key experiments, replicate measurements should be presented. Information on the outcomes of genetic analysis of the discovered clones (sequences or other information showing possible relatedness among the clones) should be included. At points, the text is not well organized (i.e. word and sentence order should be changed) and I recommend a careful re-read from all authors.
Please find below a list f remarks which I hope you will be helpful.
Line 15: „ A Fab library was built and shortened”: a Fab library was constructed and selected, do you mean?
Line 17:
The retained candidates 17 were reformatted into rabbit full IgG; thereafter, pairing tests on the recombinant strains and native 18 lysate samples were performed. – this sentence belongs to the line 21, as it starts describing the reformatting to soluble antibodies.
Line 19: I am quite sure this were more than 30 CFU – consider leaving the titre and report the size of the library after recloning (the number of independent library members).
Line 20: „enrichment by…“ – for the abstract, comment on the functional enrichment is sufficient, and enrichments are usually reported in fold-numbers.
Line 23: monoclonal antibodies are usually abbreviated as mAbs, please correct throughout the text
Line 43: “developing countries” – I believe “economically deprived” is a better expression
Line 51: “have larger numbers of neutralizing existing mabs” – interact with a larger number…, or similar
Line 69: scFvs is the usual abbreviation.
Line 73: in vitro in italics
Line 104: proprietary Fab: please include a reference
Lines 142-148: please include more experimental details, if all of this information is contained within the reference 18, please mention it at the beginning of the paragraph.
Line 174: were set-up
Line 183: for consistency with the paragraph on reformatting, here the primers should be specified as well. Alternatively, all should be summarized in a table.
Line 206: please specify the vector and the sequence used
Line 223: “on a1 mL-HiTrap”
Line 227: SDS-PAGE
Line 237: “PBS BSA 1% and sucrose 10%.”- 1% BSA-PBS and 10% sucrose, this goes for concentration labels throughout the text
Line 239: please specify the wash conditions (buffer and volume) for throughout the text
Line 241: surely this was a streptavidine-HRP conjugate? Please specify.
Line 243: please specify the percentage of the reagent.
Line 255: were prepared in PBS
Figure 1. The Figure misses error bars.
Line 279: “After a purification step on agarose following the kit protocol for Nucleospin PCR and Clean-up” – after an agarose extraction step using…
Line 282: after electrophoresis
Figure 2: The strategy and rationale behind rabbit-mouse chimeric should be briefly explained.
Line 303: Library size is in absolute numbers, and not per mL. How many colonies in total after the transformation obtained?
Lines 311-314: please communicate actual titer numbers instead of factors. What precisely does “this factor was equal to our standard” mean?
Tables 3 and 4 could be combined into one, together with the information on the sequence (if possible) and family designation. This would help in the assessment of possible clonotypic relatedness.
Line 351: and 10 different clones were expressed.
line 352: “obtained for both of the recombinant strains” – measured as the reactivity with both of the recombinant strains
Line 359: “After the engineering steps using molecular biology to reformat the Fabs into” – should be after reformatting the Fabs into”
Line 363: in CHO cells
Line 364: is this µg/ml culture?
Line 371: theoretically near 140–143 kDa – should be near the theoretical 140-143 kDa.
From Line 382: these passages are difficult to understand. I propose you shortly repeat how the ELISA outlay was, mention that titrations were performed and then summarize the criteria decisive for classification.
Figure 4: please introduce error bars or replicate experiments.
Line 444: “3.6 x 10e6 CFU/mL“ – please comment on the library size in absolute numbers (the number of independent members is important).
Line 451: please explain what is the advantage of rabbit/mouse chimeric Fab. I propose you move the sentence “Usually, a human backbone is preferred in cases of therapeutic development….” to follow the sentence: “Secondly, our library with a chimeric Fab design (rabbit/mouse) was rare as, to our knowledge, it was the first time this design was chosen and published. “
Line 470: “It was the first time we obtained this magnitude of results after an ELISA screen. “ This is great but is does not help the reader much, because we do not have the background information – please reword.
Line 475: this reference relates to a naïve library – they are very large and not pre-immune as the one described here. Please mention the differences.
Author Response
Reviewer 3
The authors describe the development of rabbit antibody pairs for diagnostic purposes, namely for sensitive detection of NP protein of hRSV. After rabbit immunization, a phage library displaying rabbit-mouse chimeric Fab was created and then selected for antigen binding. Several clones were discovered and reformatted into full-length rabbit antibodies. They then tested which antibodies work well in pairs as a capture and detection reagents, and performed a titration to discover the threshold of detection sensitivity.
The purpose and the structure of the study are clearly explained and the inference behind the experiments is clear. The presentation of data could be better (please see the remarks below). Importantly, at least for the key experiments, replicate measurements should be presented. Information on the outcomes of genetic analysis of the discovered clones (sequences or other information showing possible relatedness among the clones) should be included. At points, the text is not well organized (i.e. word and sentence order should be changed) and I recommend a careful re-read from all authors.
Please find below a list f remarks which I hope you will be helpful.
Line 15: „ A Fab library was built and shortened”: a Fab library was constructed and selected, do you mean?
This is a typo mistake; the good word is “sorted”. Change has been made in the text
Line 17:
The retained candidates 17 were reformatted into rabbit full IgG; thereafter, pairing tests on the recombinant strains and native 18 lysate samples were performed. – this sentence belongs to the line 21, as it starts describing the reformatting to soluble antibodies.
This sentence was moved after line 21 and text has been adapted.
Line 19: I am quite sure this were more than 30 CFU – consider leaving the titre and report the size of the library after recloning (the number of independent library members).
This is a typo mistake, as mentioned in results section, Library size is 3x 106 cfu/mL. correction has been made in the abstract.
Line 20: „enrichment by…“ – for the abstract, comment on the functional enrichment is sufficient, and enrichments are usually reported in fold-numbers.
As mentioned in the results section, enrichment factor is > 100. The text has been modified in the abstract.
Line 23: monoclonal antibodies are usually abbreviated as mAbs, please correct throughout the text
Changes made
Line 43: “developing countries” – I believe “economically deprived” is a better expression
Change made
Line 51: “have larger numbers of neutralizing existing mabs” – interact with a larger number…, or similar
Change made
Line 69: scFvs is the usual abbreviation.
Change made
Line 73: in vitro in italics
Change made
Line 104: proprietary Fab: please include a reference
Reference Baurand et al. originally [18] but new numbering [17] was added
Lines 142-148: please include more experimental details, if all of this information is contained within the reference 18, please mention it at the beginning of the paragraph.
The “same” protocol was replaced by the “entire” protocol in the sentence
Line 174: were set-up
Change made
Line 183: for consistency with the paragraph on reformatting, here the primers should be specified as well. Alternatively, all should be summarized in a table.
We cannot give the primer sequences use for PCR because they are confidential, and they contain information about our proprietary phagemid vector.
Line 206: please specify the vector and the sequence used
We cannot give the primer sequences use for PCR because they are confidential, and they contain information about our proprietary phagemid vector.
Line 223: “on a1 mL-HiTrap”
Change made
Line 227: SDS-PAGE
Change made
Line 237: “PBS BSA 1% and sucrose 10%.”- 1% BSA-PBS and 10% sucrose, this goes for concentration labels throughout the text
Change made
Line 239: please specify the wash conditions (buffer and volume) for throughout the text
0.05%PBS-Tween, 300 µL/w has been added to the text
Line 241: surely this was a streptavidine-HRP conjugate? Please specify.
Precision added
Line 243: please specify the percentage of the reagent.
“1 N” was added to H2SO4
Line 255: were prepared in PBS
Change made
Figure 1. The Figure misses error bars.
Duplicate measurement was performed on each serum dilution point for each date (0, 14 and 28 days). Error bars are not required.
Line 279: “After a purification step on agarose following the kit protocol for Nucleospin PCR and Clean-up” – after an agarose extraction step using…
Change made
Line 282: after electrophoresis
Change made
Figure 2: The strategy and rationale behind rabbit-mouse chimeric should be briefly explained.
Explanations are already deeply described in the Material and Method part paragraph 2.2 Library construction
Line 303: Library size is in absolute numbers, and not per mL. How many colonies in total after the transformation obtained?
We work in 5mL of transformation volume so the colony number are easy obtained by multiplied Size in cfu/mL by 5. In our present work it is 3 x 10*6 x 5 = 1.5 x 107
Lines 311-314: please communicate actual titer numbers instead of factors. What precisely does “this factor was equal to our standard” mean?
Paragraph has been rephrased
Tables 3 and 4 could be combined into one, together with the information on the sequence (if possible) and family designation. This would help in the assessment of possible clonotypic relatedness.
Scheme 3 was added with CDR3 alignments.
Line 351: and 10 different clones were expressed.
line 352: “obtained for both of the recombinant strains” – measured as the reactivity with both of the recombinant strains
Change made
Line 359: “After the engineering steps using molecular biology to reformat the Fabs into” – should be after reformatting the Fabs into”
Change made
Line 363: in CHO cells
Change made
Line 364: is this µg/ml culture?
It is a 50mL culture for pilot production; The µg/mL refers to production yield
Line 371: theoretically near 140–143 kDa – should be near the theoretical 140-143 kDa.
Change made
From Line 382: these passages are difficult to understand. I propose you shortly repeat how the ELISA outlay was, mention that titrations were performed and then summarize the criteria decisive for classification.
A conclusion section has been added to summarize all the steps.
Figure 4: please introduce error bars or replicate experiments.
Duplicate has been added in the Figure 4 legend
Line 444: “3.6 x 10e6 CFU/mL“ – please comment on the library size in absolute numbers (the number of independent members is important).
Absolute number was added in the discussion part to better compared our result with other published works.
Line 451: please explain what is the advantage of rabbit/mouse chimeric Fab. I propose you move the sentence “Usually, a human backbone is preferred in cases of therapeutic development….” to follow the sentence: “Secondly, our library with a chimeric Fab design (rabbit/mouse) was rare as, to our knowledge, it was the first time this design was chosen and published. “
Explanations are given in the end of this paragraph “we chose to work on a mouse IgG constant part as we knew that it would work well in our phage display process.” Sentence has been moved as requested.
Line 470: “It was the first time we obtained this magnitude of results after an ELISA screen. “ This is great but is does not help the reader much, because we do not have the background information – please reword.
Precision has been added in the text
Line 475: this reference relates to a naïve library – they are very large and not pre-immune as the one described here. Please mention the differences.
Naïve was added in the text